# Methylation at the C-2 position of hopanoids increases rigidity in native bacterial membranes

**Chia-Hung Wu[1], Maja Bialecka-Fornal[2], Dianne K Newman[1,3]***

[1]Division of Biology and Biological Engineering, Howard Hughes Medical Institute, California Institute of Technology, Pasadena, United States; [2]Division of Biology and Biological Engineering, California Institute of Technology, Pasadena, United States; [3]Division of Geological and Planetary Sciences, California Institute of Technology, Pasadena, United States

**Abstract** Sedimentary rocks host a vast reservoir of organic carbon, such as 2-methylhopane biomarkers, whose evolutionary significance we poorly understand. Our ability to interpret this molecular fossil record is constrained by ignorance of the function of their molecular antecedents. To gain insight into the meaning of 2-methylhopanes, we quantified the dominant (des)methylated hopanoid species in the membranes of the model hopanoid-producing bacterium *Rhodopseudomonas palustris* TIE-1. Fluorescence polarization studies of small unilamellar vesicles revealed that hopanoid 2-methylation specifically renders native bacterial membranes more rigid at concentrations that are relevant in vivo. That hopanoids differentially modify native membrane rigidity as a function of their methylation state indicates that methylation itself promotes fitness under stress. Moreover, knowing the in vivo (2Me)-hopanoid concentration range in different cell membranes, and appreciating that (2Me)-hopanoids' biophysical effects are tuned by the lipid environment, permits the design of more relevant in vitro experiments to study their physiological functions.

*For correspondence: dkn@ caltech.edu

## Introduction

Lipids play essential roles in compartmentalizing cells for specific functions and creating barriers that are selectively permeable to the environment. The composition of lipids in cell membranes varies significantly and the basic biophysical properties of membranes, such as rigidity and permeability, can be adjusted based on growth conditions and environmental stressors (*Lipowsky and Sackmann, 1995*; *Los and Murata, 2004*; Neubauer et al., 2014). One well-studied example is cholesterol in eukaryotic membranes. This essential sterol plays diverse roles in maintaining membrane structural integrity, modifying membrane rigidity, serving as a biosynthetic precursor for steroid hormones, vitamin D, and bile acids, or acting as a protein modifier for signaling pathways (*Hanukoglu, 1992*; *Gallet, 2011*; *Song et al., 2014*). In addition to their important biological functions, lipids are of interest because they are more geostable than other biomolecules. For example, hopanoid molecular fossils, 'hopanes', date back over a billion years (*Brocks et al., 2005*) and are so abundant that the global stock of hopanoids that can be extracted from sedimentary rocks is estimated to be $10^{13}$ or $10^{14}$ tons, more than the estimated $10^{12}$ tons of organic carbon in all living organisms (*Ourisson et al., 1984*). In contrast to steroids, hopanoids are a less well studied but evolutionarily significant and chemically diverse class of lipids that are thought to be sterol surrogates in bacteria (*Figure 1*) (*Rohmer et al., 1979*; *Ourisson et al., 1987*).

**eLife digest** The cell membrane that separates the inside of a cell from its outside environment is not a fixed structure. A cell can change the amount and type of different molecules in its membrane, which can alter the rigidity and permeability of the membrane and allow the cell to adapt to changing conditions.

The cell membranes of many bacteria contain molecules called hopanoids. Hopanes are the fossilized forms of these molecules and many hopanes are found extensively in sedimentary rocks. For example, 2-methylated hopanes—the fossilized forms of hopanoids that have a methyl group added to a particular carbon atom—have been found in ancient rocks that formed up to 1.6 billion years ago.

Many researchers have suggested that 2-methylated hopanes (and other molecular fossils) in sedimentary rocks could act as 'biomarkers' and be used to deduce what primitive life and ancient living conditions were like. Millions of years ago, several periods occurred where the Earth's oceans lost almost all of their oxygen; this likely placed all life on Earth under great stress. A greater proportion of the hopanes found in rocks formed during those periods are methylated than those seen in rocks from other time periods. However, it was difficult to interpret this observation about the fossil record, as the role of 2-methylated hopanoids in living bacterial cells was unknown.

Wu et al. have now investigated the role of 2-methylated hopanoids by performing experiments on bacterial membranes and found that 2-methylated hopanoids help the other molecules that make up the membrane to pack more tightly together. This makes the membrane more rigid, and the extent of this stiffening depends on the length of the 2-methylated hopanoid and on the other molecules that are present in the membrane. A more rigid membrane would protect the bacteria more in times of stress; therefore, rock layers containing an increased amount of 2-methylhopane are likely to indicate times when the bacteria living at that time were under a great deal of stress.

The rich record of ancient lipids, including fossil hopanoids, has long been recognized to hold clues into the early history of life and past environments (*Ourisson et al., 1984*; *Summons and Walter, 1990*; *Brocks and Pearson, 2005*; *Knoll et al., 2007*). But being able to confidently interpret the meaning of any ancient molecular fossil poses considerable challenges. First, we must be able to identify potential sources for these compounds, demanding unambiguous chemical parity between modern and ancient structures. Once this is achieved, understanding whether particular environmental conditions regulate the production of specific hopanoid variants becomes important. But arguably, the most critical goal in advancing our understanding of ancient lipids is being able to identify specific biological functions for their counterparts in cells today. Myriad hopanoid structures are known to exist (*Ourisson et al., 1984*), yet we only poorly understand the significance of this chemical diversity. For meaningful linkages to be made between modern compounds and ancient biomarkers, we must (1) study those hopanoids that leave a specific trace in sedimentary rocks (e.g., their chemical modifications are geostable), (2) identify their in vivo function(s), and (3) evaluate whether the roles played by these lipids in modern organisms have been conserved over the course of evolution.

Following the recognition in the early 1970s that hopanes are ubiquitous in sedimentary rocks, the occurrence of hopanoids in diverse organisms was documented, and insights were gained into their biosynthesis, biophysical properties, and cellular functions (*Ourisson et al., 1987*; *Ourisson and Rohmer, 1992*; *Pearson, 2013*). For example, studies using hopanoid-deficient mutants have shown that hopanoids promote resistance to antibiotics, detergents, extreme pH, and high osmolarity (*Welander et al., 2009*; *Sáenz, 2010*; *Schmerk et al., 2011*; *Malott et al., 2012*; *Kulkarni et al., 2013*). Biophysical studies using mixtures of hopanoids and model lipids have demonstrated that, like cholesterol, bacteriohopanetetrol cyclitol ether can condense membranes at high temperatures but fluidize membranes at low temperatures (*Poralla et al., 1980*). Similarly, bacteriohopanetetrol (BHT) and bacteriohopanemonol can condense model membranes, and diplopterol (Dip) can form liquid ordered microdomains (*Kannenberg et al., 1983*; *Nagumo et al., 1991*; *Ourisson and Rohmer, 1992*; *Sáenz et al., 2012*). A recent molecular modeling study pointed out different behaviors between Dip and BHT in their specific location within lipid bilayers and their capacity to

**Figure 1**. Structures of selected hopanoids, cholesterol, and squalene.

condense membranes, suggesting complex roles of hopanoids due to their structural diversity (*Poger and Mark, 2013*). These biophysical studies have provided insights into the physical capabilities of these hopanoids. However, whether hopanoids play the same roles in vivo has been unclear due to the differences in lipid composition and concentration between model and cellular membranes.

Among various hopanoid modifications, methylation of C-2 on the A-ring has drawn attention from Earth scientists because this modification is preserved episodically in ancient sedimentary rocks dating back to 1.6 billion years ago; accordingly, it has been suggested that 2-methylated hopanes (2Me-hopanes), the molecular fossils of 2Me-hopanoids, could potentially serve as biomarkers to interpret events in the early history of life (*Brocks et al., 2005*; *Rasmussen et al., 2008*). For a time, it was thought that 2Me-hopanes were biomarkers of cyanobacteria, and hence the process of oxygenic photosynthesis (*Summons et al., 1999*), but we now know this not to be the case (*Welander et al., 2010*; *Ricci et al., 2015*). Intriguingly, spikes in the $C_{30}$ 2Me-hopane index (ratio of methylated short hopanes to total short hopanes) through geologic time are correlated with episodes of oceanic anoxic events (OAEs), which are thought to have imposed heightened stress on the biosphere (*Knoll et al., 2007*).

Towards the goal of finding a robust interpretation for 2Me-hopanes, we have elucidated the biosynthetic pathway of 2Me-hopanoids (*Welander et al., 2010*, *2012*), linked 2Me-hopanoids to specific environments and producers through (meta)genomic studies (*Ricci et al., 2014*), and identified the stress-responsive pathway regulating transcription of the 2-methylase (*hpnP*) in the model hopanoid-producing bacterium *Rhodospeudomonas palustris* TIE-1 (*Kulkarni et al., 2013*). A major challenge in understanding the function of 2Me-hopanoids has been the lack of a clear phenotype in vivo, despite dedicated attempts to find one for the *hpnP* mutant (*Kulkarni et al., 2013*). Recent data suggest this phenotypic silence results from changes to the lipidome that compensate for the loss of 2Me-hopanoids (unpublished). To infer a specific in vivo function for 2Me-hopanoids, in vitro studies that mimic cellular composition are therefore necessary.

We hypothesized that methylation at C-2 would change hopanoids' packing with other lipids and proteins and affect membrane biophysical properties such as rigidity. Furthermore, we reasoned that such an effect would depend on the specific lipid composition of the membrane. To test these hypotheses, we took advantage of the existence of specific hopanoid-mutant strains (*Welander et al., 2010*, *2012*), and recently developed protocols allowing quantitative analysis and the purification of hopanoids and their 2-methylated species in large quantities (*Wu et al., 2015*). This experimental foundation set the stage for what we now report: in vitro membrane studies to examine how 2Me-hopanoids affect membrane rigidity in the context of different lipid environments of relevance to

the cell. Our finding that hopanoid methylation enhances membrane rigidity supports the interpretation that past intervals of heightened 2Me-hopane abundance record a history of stress.

## Results

### Whole cell membrane fluidity

To test whether 2-methylation changes membrane rigidity, we measured the membrane rigidity of specific hopanoid biosynthetic mutants (*Table 1*) (*Welander et al., 2012*) using fluorescence polarization. *Figure 2* shows the whole cell membrane rigidity measured at 25°C and 40°C. As expected, at higher temperature, the cell membrane became less rigid across all strains. At 25°C, the Δ*shc* mutant, which lacks all hopanoids, had the least rigid membrane. This could be caused by both the absence of hopanoids and the accumulation of the hopanoid biosynthetic precursor, squalene. The result is also consistent with the observation that in model lipid vesicles, hopanoids make the membrane more rigid (*Kannenberg et al., 1985*). Interestingly, the production of only short-chain hopanoids (Δ*hpnH*) is sufficient to recover the rigidity level close to that of the WT. However, when an adenosine molecule is attached to short hopanoids and accumulates in the Δ*hpnG* mutant, rigidity decreases to Δ*shc* levels. Furthermore, in Δ*hpnN* where hopanoids are unable to be transported to the outer membrane (*Doughty et al., 2011*), membrane rigidity is similar to Δ*hpnG*. These results have two implications. First, the hydrophobic reporter dye we used for fluorescence polarization measurements, diphenyl hexatriene (DPH), reflects mostly the rigidity of the outer membrane. Second, the types of hopanoids and their respective localization between the inner and outer membranes directly impact membrane rigidity.

Interestingly, no obvious impact on rigidity was found in the absence of either bacteriohopane aminotriol (Δ*hpnO*) or 2-methylhopanoid (Δ*hpnP*) or both (Δ*hpnOP*) compared to WT (*Figure 2*). This observation could have two possible explanations. One is that these specific hopanoids do not affect membrane rigidity in cells. The other explanation is that the cells might synthesize other lipids that compensate for their effects so that no phenotype is observed. We can distinguish between these two scenarios by measuring membrane rigidity using vesicles made of model lipids and purified hopanoids. However, to design experiments that are physiologically relevant, we first needed to quantify hopanoids distribution in both the inner (IM) and outer membrane (OM) of *R. palustris* TIE-1.

### Quantification of hopanoids in the inner and outer membranes of *R. palustris* TIE-1

It is well appreciated that lipids can have different subcellular localization, even in bacteria (*Matsumoto et al., 2006*). To understand the biological roles of hopanoids, especially for 2-methylhopanoids, we measured the amounts of different hopanoids in the IM and OM of *R. palustris* TIE-1 WT and Δ*hpnP* using a previously described protocol (*Figure 3*) (*Morein et al., 1994*; *Wu et al., 2015*). *Table 2* shows

**Table 1**. Mutant strains used for the whole cell membrane rigidity measurements

| Gene | Function | Deletion effect |
|------|----------|-----------------|
| *shc (hpnF)* | Cyclization of squalene to form C30 hopanoids (diploptene and diplopterol) | No hopanoid production and accumulation of squalene |
| *hpnH* | Addition of adenosine to diploptene to generate adenosylhopane, a precursor for extended hopanoid production | No extended hopanoid production, accumulation of C30 hopanoids |
| *hpnG* | Removal of adenine from adenosyl hopane | No BHT and aminoBHT production, accumulation of adenosylhopane |
| *hpnN* | An IM transporter that transports hopanoids to the outer membrane | Absence of hopanoids in the OM and accumulation of hopanoids in the IM |
| *hpnO* | Production of aminoBHT | No aminoBHT production |
| *hpnP* | Methyl transfer to A ring at C-2 | No hopanoid methylation |

The function of the gene and the effect of its deletion are listed.

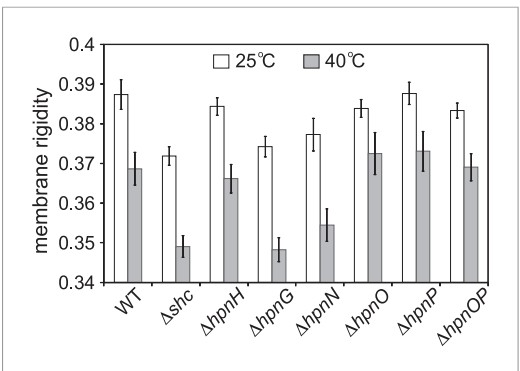

**Figure 2**. Whole cell membrane fluidity. Error bars represent the standard deviation from three biological replicates (total 21–26 technical replicates).

that the total yield of the fractionated membranes was about 10% weight of dried cells, which is comparable to 9.1% in *Escherichia coli* (*Neidhardt and Curtiss, 1996*). The yields of the total lipid extract (TLE) from the lyophilized fractionated membranes were ~12–15% from the IM and ~5% for the OM. This yield is reproducible and could be due to a larger proportion of membrane proteins in *R. palustris* TIE-1, lower recovery after lyophilization, or loss of certain lipid classes that were not quantitatively extracted by procedures optimized for hopanoids. The TLE yield in the OM was at least 50% lower than that in the IM, which is expected because the outer leaflet of the OM consists of more hydrophilic lipid A and lip-opolysaccharides that are not extractable by the hydrophobic Bligh-Dyer lipid extraction method we employed.

To quantify hopanoids, TLE from IM and OM was analyzed by GC-MS using androsterone as an internal standard and the differences in ionization efficiencies between androsterone and hopanoids were calibrated by external standards using purified hopanoids (*Figure 3*). Such calibration was recently shown to be essential for accurate hopanoid quantification (*Wu et al., 2015*). Using this approach, the exact wt% of each hopanoid in TLE was obtained. However, to put the numbers in context and compare the value in mol%, we assumed the average molecular weight of the total lipids is 786 g/mol, the same as dioleoyl phosphatidylcholine (DOPC) and *E. coli* polar lipid extract (PLE). Because we could only confidently quantify (2Me)-Dip and (2Me)-BHT, we focused our analyses on these four hopanoids. *Figure 4* shows the hopanoid quantification results. In both WT and Δ*hpnP*, each type of hopanoid is enriched in the OM compared to the IM. The total of these four hopanoids in the IM is ~2.6 mol% of TLE, whereas in the OM, the value can reach 8–11 mol%. For individual hopanoids in WT, the mol% in IM and OM are 1% and 2% for Dip, 1% and 2.4% for 2Me-Dip, 0.4% and 3.4% for BHT, and 0.1% and 0.3% for 2Me-BHT, respectively. In Δ*hpnP*, the IM and OM values are 2% and 7.5% for Dip, and 0.5% and 4.3% for BHT, respectively (*Figure 4*). This quantitative measurement of hopanoid content within the IM and OM can be used to evaluate the impact of 2-methylation upon hopanoid subcellular distribution.

The ratio of total hopanoids in the OM vs IM is 3.1 ± 0.4 and 4.4 ± 0.6 for WT and Δ*hpnP*, respectively, which is a significant difference (p = 0.038). Comparing the ratios between short ((2Me)-Dip) and long ((2Me)-BHT) chain hopanoids revealed an enrichment of long-chain hopanoids in the OM compared to the IM in both WT and Δ*hpnP* (*Figure 5*). Interestingly, about equal amounts of 2Me-Dip and Dip were found in both IM and OM, whereas 2Me-BHT is 16% and 8% of BHT in the IM and OM, respectively, suggesting hopanoid 2-methylation has neither strong nor consistent effects on the partitioning of short and long species between the IM and OM (*Figure 5*). However, our data do indicate that 2-methylation impacts that total amount of hopanoid enrichment in the OM.

## Membrane rigidity measurements using model lipid SUVs

To put these numbers in context and gain a deeper understanding on how the amount, chain-length, and 2-methylation of hopanoids impact membrane biophysical properties, we performed membrane rigidity measurements. Small unilamellar vesicles (SUVs) are commonly used for such measurements because it is straightforward to control their lipid composition (*Hope et al., 1985*). We used fluorescence polarization to measure the fluorescence of DPH, a reporter dye, in the presence of model lipids and purified hopanoids. *Figure 6* shows how the membrane rigidity of model lipids, DOPC and *E. coli* PLE, responds to the presence of cholesterol, squalene, (2Me)-Dip, and (2Me)-BHT. Because the total hopanoids in cell membranes are ~10 mol% (*Figure 4*), we varied the concentration of hopanoids between 5 and 20 mol% in SUVs, which our quantification results suggest are physiologically relevant concentrations. Starting with the simplest single lipid background, DOPC, the addition of cholesterol increases membrane rigidity and the magnitude of change is proportional to

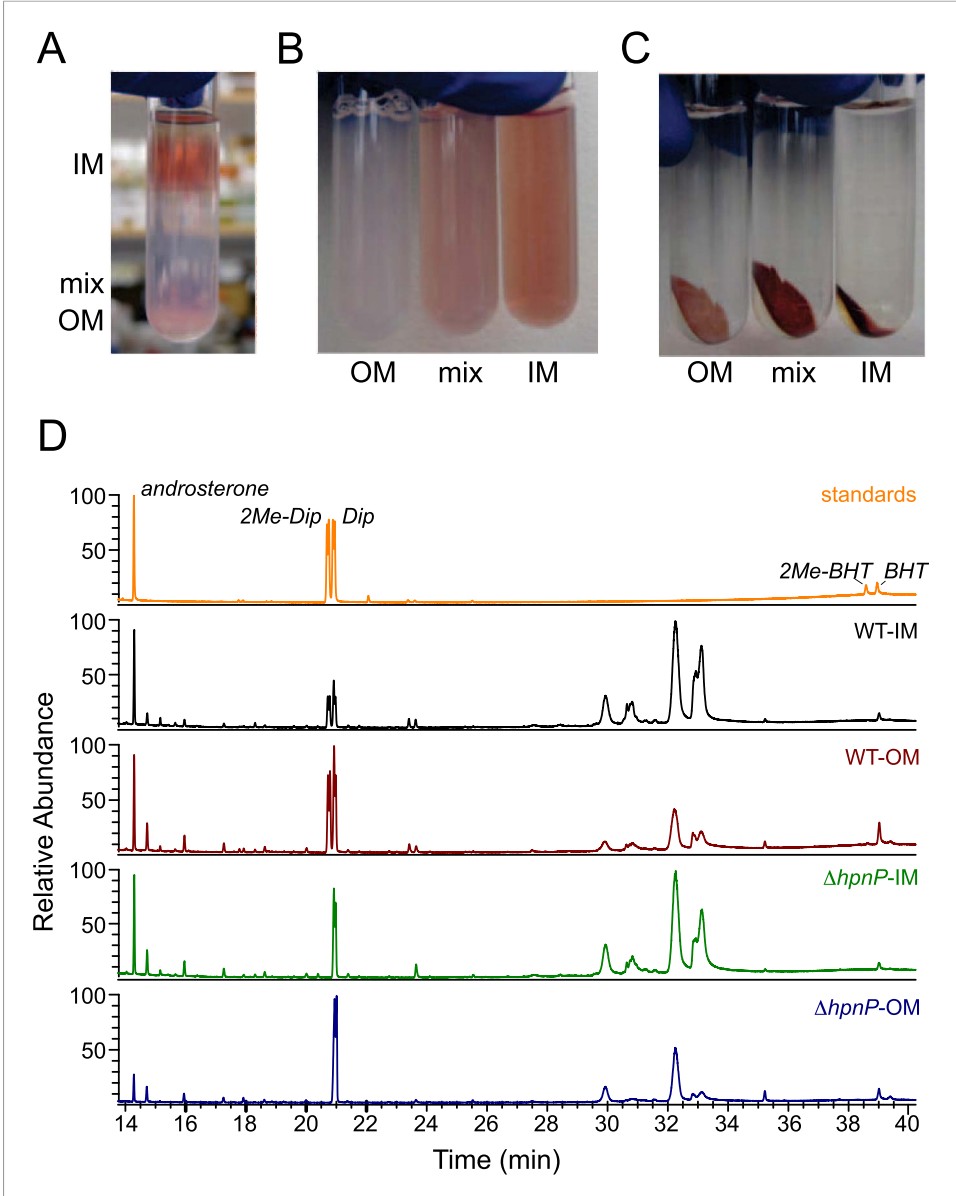

**Figure 3**. Membrane fractionation and hopanoids analysis using GC–MS. (**A**) Three distinct bands were formed after ultracentrifugation in a Percoll gradient. (**B**) The bands were recovered and resuspended. (**C**) Samples were ultracentrifuged to pellet down the purified membranes, which sat on top of a transparent solid Percoll layer. (**D**) GC–MS of fractionated membranes of *R. palustris* TIE-1 WT and Δ*hpnP*.

the amounts of cholesterol added (black line, *Figure 6A*). This observation is consistent with the literature and validates our technical approach (*Vanblitterswijk et al., 1987*). Interestingly, the addition of the hopanoid precursor, squalene, has the opposite effect as cholesterol and makes the membrane less rigid in a concentration-dependent fashion (magenta line, *Figure 6A*). However, when Dip is added, the membrane rigidity is unaffected (blue line, *Figure 6A*). Surprisingly, given the structural similarity between cholesterol and Dip, Dip does not seem to rigidify DOPC as extensively as does cholesterol. However, when Dip is further processed by the cell to produce BHT, it rigidifies DOPC vesicles in the same manner as cholesterol (red line, *Figure 6A*).

2-methylation of both Dip and BHT has striking effects on their ability to rigidify DOPC. Not only does 2-methylation increase DOPC membrane rigidity, but at 20 mol%, 2Me-Dip even outperforms cholesterol and BHT (cyan line, *Figure 6A*). 2Me-BHT also increases membrane rigidity similar to

**Table 2**. Purification yields of membrane fractionation using Percoll gradient

|  |  | Weight % membranes in dry cells | Total | Weight % TLE in membranes |
|---|---|---|---|---|
| WT | inner | 4.7 ± 0.5 |  | 12.4 ± 0.8 |
|  | mix | 3.7 ± 1.2 |  | 5.6 ± 0.1 |
|  | outer | 3.3 ± 0.4 | 11.6 ± 1.4 | 4.9 ± 0.6 |
| ΔhpnP | inner | 4.2 ± 1.3 |  | 15 ± 2.3 |
|  | mix | 2.3 ± 0.8 |  | 8.3 ± 2.3 |
|  | outer | 2.9 ± 0.6 | 9.4 ± 2.2 | 5.3 ± 1.4 |

The yields in wt% of membrane fractionation. Errors represent standard deviation from three biological replicates.

cholesterol, even though the difference from BHT is much smaller than that between Dip and 2Me-Dip (orange line, *Figure 6A*). This result demonstrates methylation itself changes the biophysical properties of hopanoids, which directly impacts membrane fluidity. This observation reinforces the interpretation that the lack of phenotype in the Δ*hpnP* strain may be due to cells synthesizing other lipids that functionally complement the absence of 2Me-hopanoids, rather than due to a lack of an impact at the molecular level per se.

To determine whether the effects on membrane rigidity by these hopanoids hold in more physiologically relevant environments, we repeated the experiment using *E. coli* PLE as the main component to form SUVs (*Figure 6B*). *E. coli* PLE from Avanti Polar Lipids consists of 67% phosphatidylethanolamine (PE), 23.2% phosphatidylglycerol (PG), and 9.8% cardiolipin (CL) (in wt%) and has an average molecular weight identical to DOPC (786 g/mol). Compared to DOPC alone, the SUVs from *E. coli* PLE are more rigid, probably because the difference in fatty acid chain saturation (green square, *Figure 6A,B*). Even though *E. coli* PLE SUVs are more rigid than DOPC SUVs to start with, we observe the same trend, where adding cholesterol rigidifies the vesicles and squalene fluidizes them. However, the concentration dependence on the rigidity change is more dramatic for cholesterol compared to squalene (black line, *Figure 6B*). Unlike in DOPC, Dip has a small rigidifying effect in the *E. coli* PLE background, yet similar to DOPC, no concentration dependence is observed (blue line, *Figure 6B*).

Addition of 2Me-Dip also rigidifies *E. coli* PLE, but in contrast to what was observed for DOPC, 2Me-Dip shows lower capacity to rigidify the membrane than cholesterol (cyan line, *Figure 6B*). Interestingly, the extended hopanoid, BHT, also rigidifies the membrane, but seemed to saturate at 10 mol% (red line, *Figure 6B*). Surprisingly, in sharp contrast to the DOPC background, 2Me-BHT strongly rigidifies the *E. coli* PLE membranes and has a concentration dependence (orange line, *Figure 6B*). This result shows that the impact of 2-methylation and hopanoid extension on a membrane biophysical property depends on the specific lipid environment. As expected, when these experiments were repeated at 40°C, membranes were less rigid overall. However, similar trends compared to 25°C were observed in both DOPC and *E. coli* PLE background (*Figure 6—figure supplement 1*).

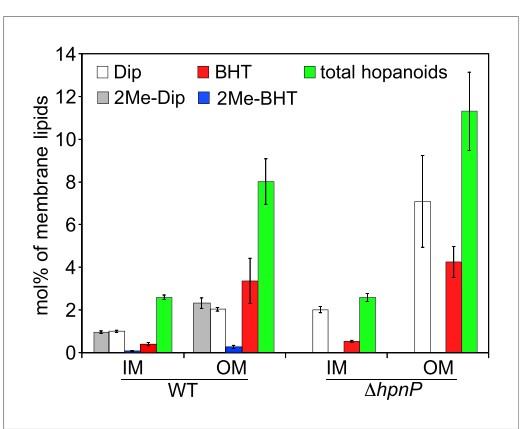

**Figure 4**. Molar percentage of hopanoids in the inner membrane (IM) and outer membrane (OM) of WT and Δ*hpnP* determined by GC–MS. Error bars represent the standard deviation from three biological replicates. Total hopanoids = sum of (2Me)-Dip and (2Me)-BHT.

## Quantification of phospholipid composition in *R. palustris* TIE-1
Given that 2-methylation of hopanoids differentially impacts membrane rigidity based on the lipid context, to understand the physiological effects of 2-methylhopanoids in the IM and OM of *R. palustris* TIE-1, we must characterize the

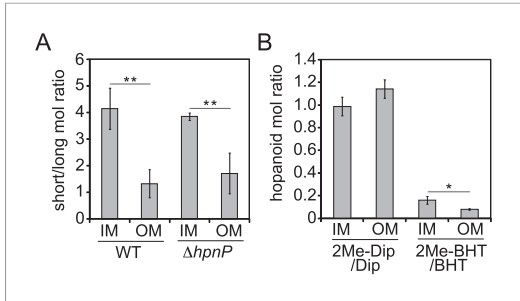

**Figure 5**. Partitioning of hopanoids in the inner membrane (IM) and outer mebraane (OM) of *R. palustris* TIE-1. (**A**) Molar ratio between short (Dip and 2Me-Dip) and long (BHT and 2Me-BHT) hopanoids in WT and ΔhpnP. (**B**) Molar ratio between methylated and desmethylated hopanoid in WT. Error bars represent the standard deviation from three biological replicates. *p = 0.015; **p < 0.01.

composition of phospholipids in native membranes and perform biophysical experiments using these membranes. To determine the exact quantity of each phospholipid in *R. palustris* TIE-1, we purified the IM and OM as described above and analyzed the lipid composition by LC-MS using electron spray ionization (*Malott et al., 2014*). The elution profiles between strains and membranes look similar (*Figure 7*). A total of 33 major phospholipids were identified, including 10 PC, 9 PE, 7 PG, and 7 CL (*Table 3*). To determine the absolute quantity of each phospholipid, exogenous standards of PC (17:0/17:0), PE (17:0/17:0), and PG (17:0/17:0) that are absent in *R. palustris* TIE-1 were added as internal standards for LC-MS analyses. Although we can detect cardiolipins, we unfortunately are unable to quantify them due to the low solubility of the cardiolipin standard in the LC-MS solvent.

*Table 4* shows the wt% of each identified phospholipid in the TLE. Among the identified phospholipids, the IM in WT and ΔhpnP has ∼50%, ∼36%, and ∼12–14% of PC, PE, and PG, respectively, whereas the OM in WT and ΔhpnP has ∼56–58%, ∼30–33%, and ∼11–12% of PC, PE, and

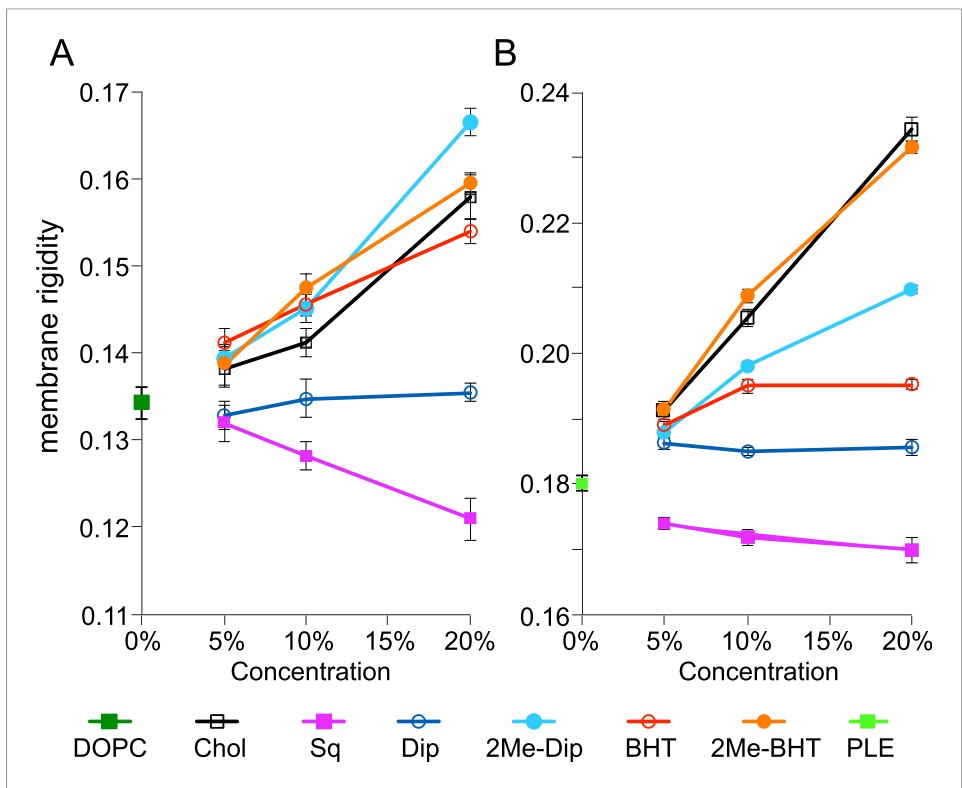

**Figure 6**. Membrane rigidity measurements at 25°C using model lipids. (**A**) Dioleoyl phosphatidylcholine (DOPC) and (**B**) *E. coli* polar lipid extract (PLE) were mixed with different mol% of cholesterol, squalene, and hopanoids. Error bars represent the standard deviation from three biological replicates (total 21 technical replicates).
The following figure supplement is available for figure 6:

**Figure supplement 1**. Membrane rigidity measurements at 25°C and 40°C using model lipids.

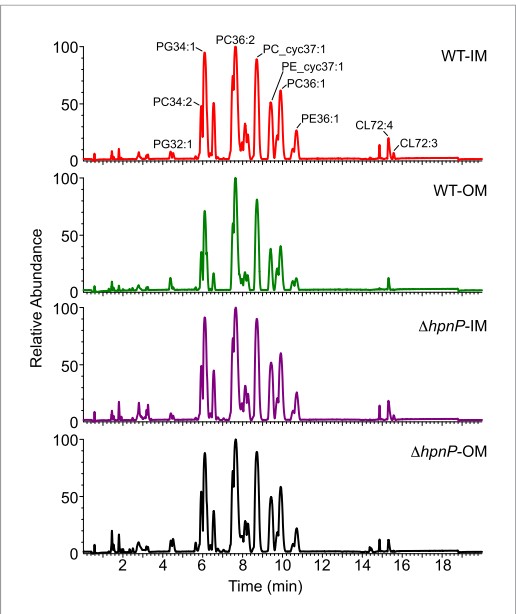

**Figure 7**. LC-MS profiles of the inner membrane (IM) and outer membrane (OM) of *R. palustris* TIE-1 WT and Δ*hpnP*.

PG, respectively. However, when we calculate the total identified phospholipid amount, it only accounts for ~26–34 wt% of the original TLE samples (*Table 4*). This low value could be due to (1) low solubility of cardiolipins and other unidentifiable lipids in LC-MS, (2) differences in the ionization efficiency of the phospholipids with different chain length or saturation (*Myers et al., 2011*), and (3) the ionization suppression effects occur from co-eluting lipids (*Brugger et al., 1997*; *Furey et al., 2013*). Nevertheless, the average molecular weight of the phospholipids without cardiolipin is 738 g/mol. Considering the higher molecular weight of cardiolipin, our calculation for the mol% of hopanoids in TLE using an average molecular weight of 786 g/mol may be very close to the real value in *R. palustris* TIE-1.

## Membrane rigidity measurements using lipid extract from *R. palustris* TIE-1 inner and outer membranes

Due to the limitations we encountered in quantifying native phospholipids, we elected to generate SUVs directly using the TLE from fractionated *R. palustris* TIE-1 IM and OM rather than reconstituting SUVs from commercially available sources. Because our main focus is on the effect of 2-methylhopanoids, we used the membranes from Δ*hpnP* and added 5, 10, and 20 mol% purified hopanoids to constitute the SUV lipid mixture. We also included WT IM and OM for comparison. *Figure 8* shows membrane rigidity measurements at 25°C using *R. palustris* TIE-1 membranes. In both WT and Δ*hpnP*, the OM showed higher membrane rigidity than IM, which could be due to the higher hopanoid content in the OM (*Figure 4*) and differences in phospholipid content. Compared to WT, the Δ*hpnP* IM and OM showed decreased rigidity. This contrasts with our whole cell membrane rigidity results in which no difference in rigidity was observed (*Figure 2*). This discrepancy could be due to the presence of lipid A and outer membrane proteins that may affect the overall membrane rigidity in whole cells.

Similar to our observations in model lipids, the addition of Dip has no effect on membrane rigidity, both in the IM and OM. However, 2Me-Dip rigidifies both IM and OM in a concentration-dependent manner, as seen in the model lipid SUVs. Interestingly, unlike the effects BHT exerts in model lipids, it does not rigidify either the IM or OM. While it is possible that BHT has no effect on native membrane rigidity, given that Δ*hpnP* membranes contain BHT, it seems more likely that endogenous BHT is saturating its membrane rigidifying capacity, similar to what we observed for *E. coli* PLE (red line, *Figure 6B*).

When BHT is methylated, it rigidifies both the IM and OM. However, different trends can be seen between these membranes: in the IM, the more 2Me-BHT present, the higher the membrane rigidity, yet OM membrane rigidity appears to saturate by 5 mol% 2Me-BHT and remains constant between 5–20 mol% of 2Me-BHT (*Figure 8B*). It is tempting to speculate that the reason less 2-methylation occurs for BHT than Dip in the OM (*Figures 4, 5*) is because less methylation of BHT is needed to significantly impact OM rigidity.

We repeated these experiments at 40°C and observed similar trends as seen at 25°C (*Figure 8—figure supplement 1*). However, we had larger standard deviations than in our model lipid experiments, which could be due to higher heterogeneity in the samples extracted from IM and OM. Nevertheless, we find clear physiologically relevant distinctions in the rigidifying effects of both short and long 2Me-hopanoids on the IM and OM of *R. palustris* TIE-1.

## Discussion

Until now, a specific role for 2Me-hopanoids in living cells has evaded experimental detection, yet its identification has been of great interest for interpreting the extensive

**Table 3**. Annotation of phospholipids identified by LC-MS analyses (see *Figure 7*)

| Compound | RT (min) | [M+H]$^+$ | [M−C$_3$H$_7$O$_2$HPO$_4$]$^+$ | [M+NH$_4$]$^+$ |
|---|---|---|---|---|
| PG32:1 | 4.39 | | 549.4895 | |
| PG34:2 | 4.89 | | 575.505 | |
| PC34:2 | 5.94 | 758.5694 | | |
| PG34:1 | 6.11 | | 577.5208 | |
| PG36:2 | 6.2 | | 603.5364 | |
| PE34:2 | 6.38 | 716.5225 | | |
| PC_cyc35:1 | 6.76 | 772.5851 | | |
| PG_cyc35 | 6.95 | | 591.5364 | |
| PG_cyc37:1 | 7.05 | | 617.5521 | |
| PC(35:2) | 7.06 | 772.5851 | | |
| PE_cyc35:1 | 7.28 | 730.5381 | | |
| PC34:1 | 7.51 | 760.5851 | | |
| PC36:2 | 7.65 | 786.6007 | | |
| PG(17:0/17:0) | 7.86 | | 579.5364 | |
| PG36:1 | 7.99 | | 605.5521 | |
| PE34:1 | 8.14 | 718.5381 | | |
| PE36:2 | 8.27 | 744.5538 | | |
| PC_cyc35 | 8.58 | 774.6007 | | |
| PC_cyc37:1 | 8.72 | 800.6164 | | |
| PE_cyc35 | 9.28 | 732.5538 | | |
| PE_cyc37:1 | 9.42 | 758.5694 | | |
| PC(17:0/17:0) | 9.75 | 762.6007 | | |
| PC36:1 | 9.91 | 788.6164 | | |
| PE(17:0/17:0) | 10.53 | 720.5538 | | |
| PE36:1 | 10.71 | 746.5694 | | |
| PC_cyc37 | 11.23 | 802.632 | | |
| PE_cyc37 | 12.1 | 760.5851 | | |
| PC36:0 | 12.79 | 790.634 | | |
| PE36 | 12.98 | 748.5851 | | |
| PC36:4 | 13.39 | 782.569 | | |
| CL70:4 | 15.09 | | | 1447.0373 |
| CL68:3 | 15.1 | | | 1421.0217 |
| CL72:4 | 15.31 | | | 1475.0686 |
| CL70:3 | 15.34 | | | 1449.053 |
| CL68:2 | 15.35 | | | 1423.0373 |
| CL72:3 | 15.58 | | | 1477.0843 |
| CL70:2 | 15.6 | | | 1451.0686 |

The types of lipids (PC: phosphatidylcholine, PE: phosphotidylethanolamine, PG: phosphatidylglycerol, CL: cardiolipin, cyc: cyclopropanation; the first number indicates the total number of carbon of the fatty acid chains and the second number indicates the number of double bonds in these chains) and their retention time (RT, min) and m/z value of the base peak are shown. For PC and PE, the base peak is the proton adduct and for CL, the base peak is the ammonium adduct. For PG, the base peak indicates a loss of glycerophosphate (−171 m/z).

**Table 4**. Phospholipid compositions in the inner membrane (IM) and outer membrane (OM) of *R. palustris* TIE-1 WT and Δ*hpnP* analyzed by LC-MS

| Compound | RT (min) | Weight % of TLE | | | |
|---|---|---|---|---|---|
| | | WT IM | WT OM | Δ*hpnP* IM | Δ*hpnP* OM |
| PC36:2 | 7.65 | 5.15 ± 0.47 | 5.34 ± 0.26 | 5.44 ± 0.57 | 5.94 ± 0.48 |
| PC_cyc37:1 | 8.72 | 4.33 ± 0.45 | 4.01 ± 0.48 | 4.07 ± 1.13 | 3.96 ± 1.24 |
| PC36:1 | 9.91 | 2.90 ± 0.3 | 1.84 ± 0.52 | 2.59 ± 0.93 | 2.15 ± 1.08 |
| PC34:1 | 7.51 | 2.66 ± 0.22 | 2.23 ± 0.33 | 2.53 ± 0.58 | 2.38 ± 0.57 |
| PC34:2 | 5.94 | 1.56 ± 0.12 | 1.24 ± 0.44 | 1.41 ± 0.51 | 1.44 ± 0.48 |
| PC_cyc35 | 8.58 | 0.23 ± 0.01 | 0.14 ± 0.05 | 0.20 ± 0.07 | 0.17 ± 0.1 |
| PC_cyc35:1 | 6.76 | 0.12 ± 0.01 | 0.07 ± 0.02 | 0.11 ± 0.04 | 0.09 ± 0.06 |
| PC(35:2) | 7.06 | 0.09 ± 0.01 | 0.05 ± 0.03 | 0.08 ± 0.03 | 0.06 ± 0.05 |
| PC_cyc37 | 11.23 | 0.07 ± 0.01 | 0.04 ± 0.01 | 0.06 ± 0.02 | 0.05 ± 0.03 |
| PC36:0 | 12.79 | 0.02 ± 0 | 0.01 ± 0 | 0.02 ± 0.01 | 0.01 ± 0.01 |
| | Sum | **17.12** | **14.96** | **16.50** | **16.25** |
| PE_cyc37:1 | 9.42 | 4.92 ± 0.36 | 3.36 ± 0.94 | 4.68 ± 1.5 | 4.18 ± 1.64 |
| PE34:1 | 8.14 | 2.49 ± 0.19 | 1.44 ± 0.49 | 2.28 ± 0.84 | 1.82 ± 0.92 |
| PE36:1 | 10.71 | 2.29 ± 0.28 | 1.08 ± 0.38 | 2.05 ± 0.85 | 1.43 ± 0.95 |
| PE36:2 | 8.27 | 1.88 ± 0.21 | 1.22 ± 0.46 | 1.74 ± 0.65 | 1.59 ± 0.72 |
| PE34:2 | 6.38 | 0.43 ± 0.02 | 0.23 ± 0.08 | 0.42 ± 0.16 | 0.33 ± 0.21 |
| PE_cyc35 | 9.28 | 0.20 ± 0.02 | 0.11 ± 0.04 | 0.19 ± 0.07 | 0.15 ± 0.09 |
| PE_cyc35:1 | 7.28 | 0.12 ± 0.01 | 0.06 ± 0.02 | 0.11 ± 0.05 | 0.09 ± 0.06 |
| PE_cyc37 | 12.1 | 0.07 ± 0.01 | 0.04 ± 0.01 | 0.07 ± 0.02 | 0.05 ± 0.03 |
| PE36 | 12.98 | 0.04 ± 0 | 0.03 ± 0.01 | 0.04 ± 0.01 | 0.03 ± 0.01 |
| | Sum | **12.44** | **7.58** | **11.58** | **9.67** |
| PG36:2 | 6.2 | 2.59 ± 0.24 | 1.68 ± 0.54 | 2.25 ± 0.67 | 1.87 ± 0.8 |
| PG36:1 | 7.99 | 1.13 ± 0.08 | 0.69 ± 0.19 | 0.98 ± 0.31 | 0.81 ± 0.35 |
| PG34:1 | 6.11 | 0.77 ± 0.17 | 0.5 ± 0.2 | 0.6 ± 0.24 | 0.51 ± 0.19 |
| PG_cyc37:1 | 7.05 | 0.11 ± 0.02 | 0.07 ± 0.05 | 0.08 ± 0.02 | 0.07 ± 0.04 |
| PG34:2 | 4.89 | 0.05 ± 0 | 0.03 ± 0.01 | 0.04 ± 0.01 | 0.03 ± 0.02 |
| PG_cyc35 | 6.95 | 0.02 ± 0 | 0.01 ± 0 | 0.02 ± 0.01 | 0.01 ± 0.01 |
| PG32:1 | 4.39 | 0.00 ± 0 | 0.01 ± 0 | 0 ± 0 | 0.01 ± 0 |
| | Sum | **4.67** | **3** | **3.97** | **3.31** |
| **PC + PE + PG** | **Total % of TLE** | **34.23** | **25.54** | **32.06** | **29.23** |

2Me-hopane fossil record (*Welander et al., 2010*; *Kulkarni et al., 2013*). Our findings that 2Me-hopanoids rigidify membranes to different extents depending both on their specific structure (short or long) and lipid context not only provide a clear biological function for these compounds, but also help rationalize why previous efforts to identify such a function have been challenging. That methylation per se can contribute to rigidifying membranes may also help explain the association of methylated hopanoids in certain modern and ancient environments.

Under what circumstances would adding a methyl group at the 2′ position of hopanoids, which seems a rather small modification, be beneficial? Might there be a mechanistic explanation for the enrichment of 2Me-hopanes during stressful OAEs (*Knoll et al., 2007*)? Several independent lines of evidence bridge our biophysical findings with the abundance patterns in the rock record, together suggesting that 2-methylhopanoids confer stress resistance: (1) 2Me-hopanoids are enriched in the

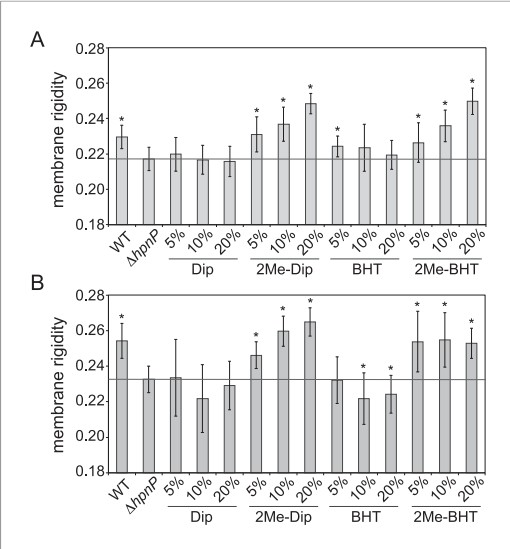

**Figure 8.** Membrane rigidity measurements at 25°C using total lipid extract from *R. palustris* TIE-1 inner membrane (IM) and outer membrane (OM). The IM (**A**) or OM (**B**) from Δ*hpnP* was mixed with different mol% of hopanoids. Error bars represent the standard deviation from three biological replicates (total 21 technical replicates). *p < 0.001 (relative to Δ*hpnP*).

The following figure supplement is available for figure 8:

**Figure supplement 1**. Membrane rigidity measurements at 40°C using total lipid extract from *R. palustris* TIE-1 inner membrane (IM) and outer membrane (OM).

outer membrane of akinetes, survival cell types of the cyanobacterium *Nostoc punctiforme* (*Doughty et al., 2009*), (2) a stress-responsive pathway upregulates the HpnP methylase in *R. palusris* (*Kulkarni et al., 2013*), (3) in modern environments, the capacity for 2Me-hopanoid production significantly correlates with organisms, metabolisms (e.g., nitrogen fixation), and environments that support plant–microbe interactions (*Ricci et al., 2014*). This correlation, together with the observation that (2Me)-hopanoids promote symbiosis (*Silipo et al., 2014*), tempt us to speculate that 2Me-hopanoids may indirectly facilitate nitrogen fixation by enhancing bacterial survival under the stressful conditions that accompany the establishment of symbiosis (*Gibson et al., 2008*). Going forward, it is worth critically examining this hypothesis; if correct, such an interpretation would indicate that spikes in the 2Me-hopane index may reflect episodes of particular environmental stresses favoring the growth of organisms capable of withstanding it using (2Me)-hopanoids.

How might the 2-methylation of hopanoids permit such an adventitious adaptation? Molecular dynamic simulations of 2Me-hopanoid within a relevant lipid context are required to understand the interactions on an atomic scale. However, we may speculate about the mechanism of rigidification through lessons learned from studies of cholesterol, in which the number of methyl groups control its optimal molecular packing to geometrically complement phospholipid chains (*Bloch, 1979*). The addition or removal of methyl groups over the evolution of the cholesterol lipid family is thought to have optimally tuned cholesterol's ability to order or condense phospholipid membranes (*Miao et al., 2002*; *Rog et al., 2007*). We suggest that when a methyl group is added onto the 2′ position of A-ring in hopanoid, the stearic hindrance between the 2-methyl group and the methyl groups at the 4′ and 10′ position of the A-ring could transform the ring from a chair to a twisted conformation. The two additional 1,3-diaxial interactions elicited by hopanoid 2-methylation could mimic the smoothing and/or tilting effect known for cholesterol, thus rationalizing how 2-methylation may improve the ability of hopanoids to rigidify membranes (*Rog et al., 2007*).

The differences in (2Me)-hopanoid distribution between the IM and OM pose many interesting questions about their role in maintaining membrane integrity and homeostasis. Compared to (2Me)-Dip, the addition of a hydrophilic tail to form (2Me)-BHT or even BHT-glucosamine (*Figure 1*) may favor a stronger interaction with the outer leaflet of the OM, in which the lipid head group is heavily modified with hydrophilic molecules. The relative enrichment of (2Me)-BHT in the OM is consistent with such a scenario. The hydrophilic tail of 2Me-BHT could also affect the vertical position of the 2-methyl group in the membrane compared to 2Me-Dip (*Rog et al., 2007*; *Poger and Mark, 2013*), which may explain the difference between 2Me-BHT and 2Me-Dip in rigidifying membranes with different compositions. Future research will illuminate whether there are additional interactions between hopanoids and other membrane constituents (e.g., proteins or cell wall components) that facilitate survival under stress.

Finally, it is important to keep in mind that (2Me)-hopanoids may act locally rather than globally with respect to influencing membrane biophysical properties. In our *in vitro* experiments, we did not observe a significant difference in membrane rigidity when less than 10 mol% of (2Me)-hopanoids were used (*Figure 6*). This may be due to a critical concentration needed to trigger an effect, which is consistent with molecular dynamic simulations that demonstrate cholesterol starts to self-organize within membranes at concentration above 10 mol% (*Martinez-Seara et al., 2010*). However, we hasten to point out that the mol% of (2Me)-hopanoids in the native membrane experiment (*Figure 8*) are not directly comparable to those using model lipids (*Figure 6*) due to the presence of endogenous hopanoids in *ΔhpnP* membranes. The existence of BHT in *ΔhpnP* may explain why the membrane rigidifying effect of exogenously added BHT is saturated at 10 mol% in the *E. coli* PLE but has no impact on *R. palustris* TIE-1 IM and OM. In this context, it is noteworthy that cardiolipin significantly increases in the absence of all hopanoids (unpublished). In *E. coli*, cardiolipin localizes to negatively curved regions of the cell (*Renner and Weibel, 2011*). Looking beyond the function of methylation, it is possible that certain hopanoid types could fulfill the geometry requirements of curved membranes and facilitate cell division or vesicle formation, consistent with both the microdomain features observed in prior subcellular hopanoid localization studies (*Doughty et al., 2014*) and the strong cell division defect displayed by a mutant lacking the ability to transport hopanoids to the OM (*Doughty et al., 2014*). Going forward, consideration of other roles for structurally diverse hopanoids, including the possibility that some might influence membrane protein function (*Phillips et al., 2009*), modify specific proteins or cell wall components through covalent linkages (*Jeong and McMahon, 2002*; *Silipo et al., 2014*), or even play a role in signaling pathways in analogy to cholesterol and phosphatidylcholine (*Kuwabara and Labouesse, 2002*; *Aktas et al., 2010*), will enhance our appreciation for this ancient lipid class.

## Materials and methods

### Bacterial strains and chemicals

*R. palustris* TIE-1 wild type (WT) and mutant strains were grown as previously described (*Welander et al., 2012*). Purified hopanoids ((2Me)-diplopterol, (2Me)-bacteriohopanetetrol [BHT]) were obtained by following the purification protocols (*Wu et al., 2015*). *E. coli* polar lipid extract and 1,2-dioleoyl-*sn*-glycero-3-phosphocholine (DOPC), 1,2-diheptadecanoyl-*sn*-glycero-3-phosphocholine (PC[17:0/17:0]), 1,2-diheptadecanoyl-*sn*-glycero-3-phosphoethanolamine (PE[17:0/17:0]), 1,2-diheptadecanoyl-*sn*-glycero-3-phospho-(1′-*rac*-glycerol) (PG[17:0/17:0]) were from Avanti Polar Lipids (Alabaster, AL). Squalene, cholesterol, pyridine, acetic anhydride, morpholinepropanesulfic acid (MOPS), 4-(2-hydroxyethyl)-1-piperazineethanesulfonic acid (HEPES), sodium succinate, 1,6-diphenyl-1,3,5-hexatriene (DPH), Percoll, and tetrahydrofuran (THF) were from Sigma–Aldrich (Milwaukee, WI). Yeast extract was from HIMEDIA (Mumbai, India). Peptone was from BD Biosciences (San Jose, CA). Methanol and dichloromethane (DCM) were HPLC grade from Alfa Aesar (Ward Hill, MA).

### Whole cell membrane fluidity measurements

To prepare bacterial cells for measurements of membrane fluidity, single colonies of *R. palustris* TIE-1 WT and mutants were inoculated into 10 ml YPMS (0.3% yeast extract, 0.3% peptone, 50 mM MOPS, 5 mM succinate, pH 7.0) and grown at 30°C, 250 rpm for ~72 hr to reach late stationary phase ($OD_{600}$ ~1.0). The cells (250 µl) were spun down and the cell pellets were washed once with 50 mM HEPES, 50 mM NaCl, pH 7.0 (buffer A). Pellets were resuspended in different amounts of the same buffer to adjust the final $OD_{600}$ to ~0.2. To measure membrane fluidity, 4.3 µl of DPH (736 µM stock solution in ethanol; the concentration was determined by $\varepsilon 350\ nm = 88\ cm^{-1}\ mM^{-1}$ in methanol) was added into 400 µl of the cell suspension. Samples were incubated in a 25°C or 40°C water bath without light for 30 min, followed by measurements of fluorescence polarization (Fluorolog, HORIBA Instruments (Edison, NJ). Instrument parameter: ex 358 nm, slit = 3 mm; em 428 nm, slit = 7 mm; integration time = 1 s) (*Lin et al., 2011*). Three biological replicates were measured, each containing 6–14 technical replicates. To reduce bias from the stability of the instrument and the samples, especially at 40°C, we randomized our data acquisition sequence. p value in this manuscript represents *t*-test using two-tailed equal variance.

## Membrane fractionation using Percoll gradient

To prepare cell cultures for membrane fractionation, single colonies of *R. palustris* TIE-1 WT or Δ*hpnP* mutant were inoculated into 10 ml YPMS and grown for ~4 days at 30°C, 250 rpm. The culture (0.5 ml) was then inoculated into 1 l of YPMS in a 2-l flask and grown at 30°C, 250 rpm for 4 days before harvesting by centrifugation at 12,000×g for 20 min at RT. The typical yield was ~1.8 g of wet cell paste per 1 l culture. To estimate the yield in dried cells, a small aliquot of the wet cell paste was lyophilized until there was no further change in weight. On average the weight of dried cells was one third of wet cells. The wet cell pastes were stored at −80°C before cell lysis.

To lyse the cells, 19 ml of buffer A was added into ~3.6 g of cells (from 2 l culture) and passed through a French Press twice at 14,000 psi, followed by sonication (Sonic Dismembrator 550, Fisher Scientific (Waltham, MA), 1/8 inch tip, power output 3.5, 1 s on, 4 s off, total on time 5 min at 4°C). The cell debris was spun down at 20,000×g, 20 min at 4°C. The supernatant containing cell membranes was transferred into 4-ml ultracentrifugation tubes (~3 ml sample per tube) and centrifuged at 80,000 rpm in a TLA-100.3 rotor for 1 hr at 4°C (Optima MAX Ultracentrifuge, Beckman Coulter (Brea, CA)). The resulting membrane pellets in each tube were resuspended in 300 µl buffer A by pipetting while being sonicated in a bath sonicator (VWR (Radnor, PA) B2500A-DTH, 42 kHz, RF Power 85 W). The suspension was combined into one single tube and sonicated again using the probe sonicator (power level 2.5, 1 s on, 4 s off, total on time 2.5 min, 4°C).

To separate inner and outer membranes, ~320 µl of membrane samples were laid on top of 3 ml 18% Percoll (vol/vol in buffer A), followed by ultracentrifugation at 30,000 rpm in a TLA 100.3 rotor at 4°C for 15 min (*Morein et al., 1994*). Three visually distinct bands were formed and a pipetman was used to take in sequence of the top band (1 ml), bottom band (~200–250 µl), and the middle band (0.7–1 ml) (*Figure 3*). The top and bottom bands constituted the IM and OM, determined by the presence and absence of NADH-oxidase activity, respectively. The band on top of the OM was less defined and exhibited some NADH-oxidase activity, which may be an artifact from sonication steps that mixed the IM and OM, and we therefore discarded it in our subsequent studies. To remove Percoll, samples from the same band were combined and centrifuged in a TLA 100.3 rotor at 50,000 rpm at 4°C for 1.5 hr. After centrifugation, a layer of the fractionated membrane was formed on top of a transparent Percoll layer (*Figure 3*). The membrane layers were collected by pipetting gently in water and/or scraped gently using a metal spatula. The membrane samples were then frozen at −20°C before being lyophilized. The total lipid extractions (TLE) from the lyophilized membranes were obtained by modified Bligh–Dyer extraction according to published protocols (*Kulkarni et al., 2013*).

## Quantification of lipid compositions of inner and outer membranes of *R. palustris* TIE-1

GC-MS (Thermo Scientific (Waltham, MA) Trace-GC/ISQ mass spectrometer with a Restek Rxi-XLB column [30 m × 0.25 mm × 0.10 µm]) was used to quantify hopanoids from the TLE from the IM and OM. An internal standard, androsterone (750 ng) was air dried with 100 µg of the TLE overnight at RT and derivatized with 50 µl acetic anhydride and 50 µl pyridine at 60°C for 30 min, followed by GC-MS analyses as described (*Welander et al., 2009*; *Kulkarni et al., 2013*). To account for the difference in ionization efficiencies between androsterone and hopanoids, calibration curves using androsterone and purified hopanoids were generated to quantify hopanoids (*Wu et al., 2015*).

LC-MS (Waters (Milford, MA) Acquity UPLC/Xevo G2-S time-of-flight mass spectrometer with a CSH C18 column [2.1 × 100 mm × 1.7 µm]) was used to quantify both phospholipids and hopanoids. Phospholipid internal standards (PC[17:0/17:0], PE[17:0/17:0], PG[17:0/17:0], 1 µg each) were mixed with 100 µg of the TLE from fractionated membranes and air-dried overnight at RT. LC solvent (200 µl, Isopropanol:acetonitrile:water = 2:1:1) was then added into the samples, followed by sonication before analyses by LC-MS as described earlier (*Malott et al., 2014*; *Wu et al., 2015*). The column temperature was maintained at 55°C. A binary solvent system containing solvent A (acetonitrile:water; 60:40) and solvent B (isopropanol:acetonitrile; 90:10), both with 10 mM ammonium formate and 0.1% formic acid was used. The flow rate was set at 400 µl/min and the elution program started at 40% B, increased linearly to 43% B in 2 min, then to 50% B in 0.1 min, followed by a linear increase to 54% B over 9.9 min, a jump to 70% B in 0.1 min, another linear

increase to 99% B over 5.9 min, a subsequent decrease to 40% B in 0.1 min, and then maintained at the same level for 1.9 min.

The eluents from the column were ionized by electrospray ionization (ESI). $MS^E$ data from 100 to 1500 m/z were collected in either the positive or negative ion mode. $MS^E$ consisting of both low energy and high energy scans were obtained simultaneously. During data analysis product ions can be associated with parent ions if they are coincident in chromatographic time. Electrospray conditions were capillary voltage 2.0 kV, cone voltage 30 V, source offset 60 V, source temperature 120°C, desolvation temperature 550°C, cone gas 20 l/hr, and desolvation gas 900 l/hr. The TOF-MS was run in resolution mode, typically 32,000 m/Δm. The mass axis was calibrated with sodium formate clusters. Leucine enkephalin was used as a mass reference during acquisition. The data were collected in continuum mode, and then converted to centroid mode for quantitative analysis using the Quanlynx program (Waters Corporation, Milford, MA) (*Wu et al., 2015*).

## Membrane fluidity measurements in small unilamellar vesicles (SUVs)

Cholesterol, squalene, and hopanoid stock solutions were prepared at 1 mg/ml in THF and the *E. coli* PLE and DOPC were prepared at 10 mg/ml in DCM. To prepare lipid mixture, a total of 1 μmol of lipid was added into 0.5 ml of DCM and dried in a rotary evaporator. Because any residual solvents can cause high errors in fluidity measurements, the samples were placed under vacuum overnight to ensure complete removal of organic solvents.

To prepare SUVs, 1 ml of buffer A was added into the glass vials containing dried lipid mixtures. Samples were suspended by sonication for 1 hr at RT in a bath sonicator (VWR B2500A-DTH, 42 kHz, RF Power 85 W). The suspended lipids (murky giant multilamellar vesicles) were transferred into 1.5-ml eppendorf and flash frozen in liquid nitrogen for 3 min, followed by thawing in a 37°C water bath for 3 min. This freeze–thaw cycle, which breaks down the giant vesicles into smaller ones, was repeated two more times. SUVs were prepared by passing the samples through 0.1-μm polycarbonate membranes (Whatman) using Avanti mini-Extruder at RT (Avanti Polar Lipids). The extrusion was performed a total of 11 times and the vesicle suspension became clear during the process. The sizes and stability of the SUVs was determined by dynamic light scattering (Wyatt (Santa Barbara, CA) DynaPro NanoStar. Instrument parameters: acquisition time 5 s, number of acquisition 10, laser wavelength 659 nm, laser power 10%, 25°C). The average size distribution of the SUVs was between 80 and 90 nm and remained stable for at least 4 hr at RT.

SUVs after extrusion were diluted 1:1 in buffer A to reach a final concentration of 0.5 mM (400 μl total volume). DPH (1.8 μl of 44.5 μM stock solution in ethanol) was added into the sample and vortexed immediately. The SUV-DPH samples were incubated in 25°C or 40°C water bath without light for at least 30 min before the fluorescence polarization was measured using the parameters described above. The concentration of the fluorescence reporter dye DPH and the instrument parameters were optimized to have a strong and linear signal output.

## Membrane fluidity measurements using total lipid extract from inner and outer membranes of *R. palustris* TIE-1

Different amounts of purified hopanoids (5, 10, and 20 nmol) were added to 100 nmol of the inner or outer membrane extracts of Δ*hpnP* (assuming average molecular weight is 786 g/mol). The same procedures as described above were followed for the preparation of SUVs. Buffer A (600 μl) was used to suspend the dried lipids so that the final lipid concentrations before the addition of DPH were between 0.088 and 0.1 mM (400 μl sample volume). To measure fluorescence polarization, DPH (1.8 μl of 7.4 μM stock solution in ethanol) was used (the final concentration of DPH was 0.03 μM, which was ~0.03 mol% of the total lipids in the sample). Controls of membranes from WT or Δ*hpnP* only without addition of hopanoids were included.

## Acknowledgements

We thank the members of the Newman lab for critical comments on the manuscript. We thank Dr Nathan Dalleska for help with UPLC-TOF-MS. The UPLC-TOF-MS equipment in the California Institute of Technology's Environmental Analysis Center was used in the work. We thank Dr Heun Jin Lee and Dr Eva Schmid for vesicle preparation advice. This work was supported by grants from NASA

(NNX12AD93G), the National Science Foundation (1224158), and the Howard Hughes Medical Institute (HHMI) to DKN. DKN is an HHMI Investigator.

## Additional information

### Funding

| Funder | Grant reference number | Author |
| --- | --- | --- |
| National Aeronautics and Space Administration (NASA) | NNX12AD93G | Chia-Hung Wu, Maja Bialecka-Fornal |
| National Science Foundation (NSF) | 1224158 | Chia-Hung Wu, Maja Bialecka-Fornal |
| Howard Hughes Medical Institute (HHMI) | | Chia-Hung Wu, Dianne K Newman |

The funders had no role in study design, data collection and interpretation, or the decision to submit the work for publication.

### Author contributions

C-HW, MB-F, Conception and design, Acquisition of data, Analysis and interpretation of data, Drafting or revising the article; DKN, Conception and design, Analysis and interpretation of data, Drafting or revising the article

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
