## [Decision Letter]

Thank you for sending your work entitled “Methylation at the C–2 position of hopanoids increases rigidity in native bacterial membranes” for consideration at *eLife*. Your article has been favorably evaluated by Richard Losick (Senior editor) and 4 reviewers, one of whom is a member of our Board of Reviewing Editors.

All the individuals responsible for the peer review of your submission have agreed to reveal their identity: Jon Clardy (Reviewing editor); Andrew Knoll (peer reviewer), Martin Burke (peer reviewer), and Alex Brown (peer reviewer).

The Reviewing editor and the other reviewers discussed their comments before we reached this decision, and the Reviewing editor has assembled the following comments to help you prepare a revised submission.

The reviewers were uniformly enthusiastic about the suitability of a revised version of this study for *eLife*. The suggested areas for revision were:

1) The study makes a valuable contribution to the wide but sporadic distribution of 2Me-hopanes in sedimentary rocks. However, the jump from experimental results to geological explication is a bit abrupt, and the motivating issue of geological occurrence should be expanded. The discussion of nitrogen fixation by *R. palustris*, and possibly other bacteria, could also be expanded.

2) Lipid quantification forms an essential part of this study, and some parts need to be clarified. The citation in the Results section to an as yet unpublished manuscript (Neubauer et al., 2014) doesn't make clear what method was used, and a brief mention of ionization mode, chromatography conditions, and other relevant information should be added. In the same section, some comments on possible sources of error for low values are mentioned. In this regard, it is known that PC, PE, and PG have very different head groups and ionize very differently in MS; PG ionizes well in negative mode, PE ionizes well in both positive and negative, and PC ionizes better in positive mode. In addition, chain length and degree of unsaturation also affect ionization. Previous work in the area of lipid analysis using mass spectrometry should be included. In particular, the well documented effects on ion suppression should be mentioned (see [5], in PNAS). General lipidomic methodologies and reviews on phospholipids (H. A. Brown), neutral lipids (Robert Murphy), and sterols (David Russell) might be useful.

3) The manuscript clearly establishes that the introduction of a methyl group at C–2 of the hopane skeleton has a dramatic effect on membrane rigidity. The speculation that the structural basis of the effect could arise from the twisting of the A–ring from chair to boat is fundamentally sound, and as the manuscript notes, forms a promising area for future work. The authors might want to mention that, although 1,3-diaxial interactions of the C4 and C6 methyl groups in sterols (e.g. lanosterol) do not appear to distort the A–ring, in the context of the hopane scaffold (lacking an equatorial C3 hydroxyl) these interactions themselves could be sufficient to distort the A–ring into a twist boat and/or the methyl at C–2 would alter packing by changing interactions in the plane of the hopane structure.

---

## [Author Response]

*1) The study makes a valuable contribution to the wide but sporadic distribution of 2Me-hopanes in sedimentary rocks. However, the jump from experimental results to geological explication is a bit abrupt, and the motivating issue of geological occurrence should be expanded. The discussion of nitrogen fixation by* R. palustris*, and possibly other bacteria, could also be expanded*.

We have significantly revised the Introduction and Discussion to make the linkages between the geological context for this work and the significance of our biophysical discoveries more clear.

*2) Lipid quantification forms an essential part of this study, and some parts need to be clarified. The citation in the Results section to an as yet unpublished manuscript (**Neubauer et al., 2014**) doesn't make clear what method was used, and a brief mention of ionization mode, chromatography conditions, and other relevant information should be added. In the same section, some comments on possible sources of error for low values are mentioned. In this regard, it is known that PC, PE, and PG have very different head groups and ionize very differently in MS; PG ionizes well in negative mode, PE ionizes well in both positive and negative, and PC ionizes better in positive mode. In addition, chain length and degree of unsaturation also affect ionization. Previous work in the area of lipid analysis using mass spectrometry should be included. In particular, the well documented effects on ion suppression should be mentioned (see*
[5]*, in PNAS). General lipidomic methodologies and reviews on phospholipids (H. A. Brown), neutral lipids (Robert Murphy), and sterols (David Russell) might be useful*.

Detailed LC-MS conditions have been added into the Materials and methods section. We also rephrased our discussion of the possible cause for the low calculated quantities of phospholipids for clarity. Additional references have been added.

*3) The manuscript clearly establishes that the introduction of a methyl group at C–2 of the hopane skeleton has a dramatic effect on membrane rigidity. The speculation that the structural basis of the effect could arise from the twisting of the A–ring from chair to boat is fundamentally sound, and as the manuscript notes, forms a promising area for future work. The authors might want to mention that, although 1,3-diaxial interactions of the C4 and C6 methyl groups in sterols (e.g. lanosterol) do not appear to distort the A–ring, in the context of the hopane scaffold (lacking an equatorial C3 hydroxyl) these interactions themselves could be sufficient to distort the A–ring into a twist boat and/or the methyl at C–2 would alter packing by changing interactions in the plane of the hopane structure*.

We have modified the text according to the reviewers’ comments.